# Green Extraction of Bioactive Compounds from Plant Biomass and Their Application in Meat as Natural Antioxidant

**DOI:** 10.3390/antiox10091465

**Published:** 2021-09-15

**Authors:** Alzaidi Mohammed Awad, Pavan Kumar, Mohammad Rashedi Ismail-Fitry, Shokri Jusoh, Muhamad Faris Ab Aziz, Awis Qurni Sazili

**Affiliations:** 1Institute of Tropical Agriculture and Food Security, Universiti Putra Malaysia, Seri Kembangan 43400, Malaysia; gs54136@student.upm.edu.my (A.M.A.); pavankumar@gadvasu.in (P.K.); 2Department of Livestock Products Technology, College of Veterinary Science, Guru Angad Dev Veterinary and Animal Sciences University, Ludhiana 141004, Punjab, India; 3Department of Food Technology, Faculty of Food Science and Technology, Universiti Putra Malaysia, Seri Kembangan 43400, Malaysia; ismailfitry@upm.edu.my; 4Department of Animal Science, Faculty of Agriculture, Universiti Putra Malaysia, Seri Kembangan 43400, Malaysia; shokri@upm.edu.my (S.J.); mhd_faris@upm.edu.my (M.F.A.A.)

**Keywords:** plant extracts, conventional and green extraction, bioactive compounds, antioxidant effect

## Abstract

Plant extracts are rich in various bioactive compounds exerting antioxidants effects, such as phenolics, catechins, flavonoids, quercetin, anthocyanin, tocopherol, rutin, chlorogenic acid, lycopene, caffeic acid, ferulic acid, p-coumaric acid, vitamin C, protocatechuic acid, vitamin E, carotenoids, β-carotene, myricetin, kaempferol, carnosine, zeaxanthin, sesamol, rosmarinic acid, carnosic acid, and carnosol. The extraction processing protocols such as solvent, time, temperature, and plant powder should be optimized to obtain the optimum yield with the maximum concentration of active ingredients. The application of novel green extraction technologies has improved extraction yields with a high concentration of active compounds, heat-labile compounds at a lower environmental cost, in a short duration, and with efficient utilization of the solvent. The application of various combinations of extraction technologies has proved to exert a synergistic effect or to act as an adjunct. There is a need for proper identification, segregation, and purification of the active ingredients in plant extracts for their efficient utilization in the meat industry, as natural antioxidants. The present review has critically analyzed the conventional and green extraction technologies in extracting bioactive compounds from plant biomass and their utilization in meat as natural antioxidants.

## 1. Introduction

The meat industry’s focus is shifting towards antioxidants that are novel, efficient, economical, green, or natural alternatives to potentially harmful synthetic preservatives. There has been an ever-increasing effort by the meat industry to explore novel, effective and edible antioxidants and antimicrobials compounds obtained from natural sources. Recently, there have been renewed interests in applying natural compounds (plant extracts, herbs, spices) in food preservation. Since ancient times, these additives have been added to foods as flavorings/seasonings, folk medicine, and food preservatives owing to antioxidant, bacteriostatic or bactericidal properties of polyphenolic compounds, and essential oils. The consumption of these compounds is known to exert beneficial effects on consumer’s health in addition to their preservative role. These herbs, spices, essential oils, or plant extracts exhibit various protective actions against the occurrence of diseases, and thus are widely used as potential green alternatives to food additives, preservatives, and dietary supplements.

Natural antioxidants are safe to consume, available in large quantities at a relatively lower price than their synthetic counterparts, and can be extracted and applied in the meat industry [1]. These compounds are abundantly present in the vegetable kingdom, such as in spices, herbs, and essential oils, with their concentration varying in throughout a plant, for example bark in the cinnamon and arjuna trees, the roots of liquorice, rosemary, cloves, and grape seeds, and the leaves of oregano and tea. Natural antioxidants are recognized as nutraceutical ingredients or supplements that can be used for raw meat and/or meat products [2]. The commonly used spices and herbs (such as oregano, rosemary, cinnamon, aniseed, fennel, basil, garlic, and ginger) are rich in phenolic compounds like phenolic acids, phenolic diterpenes, flavonoids, volatile oils, carnosic, caffeic, and chlorogenic acid. In their B rings, these polyphenols have 30–40 dihydroxy groups, and galloyl ester in the C rings of flavonoids can bind iron, making these compounds very potent antioxidants [3]. Phenolic compounds are lipid-soluble, owing to the hydroxyl group present in their chemical structure; they react with microbes’ cellular membranes, leading to loss of cell membrane integrity and, thus, antimicrobial effect. Fruits such as apple, plum, grape, pomegranate, and several types of berries, such as blueberries, cranberries, and bearberries, which are all good sources of antioxidants, owing to their high concentrations of flavonoids.

## 2. Plant Extracts as Natural Antioxidants

At present, various plant extracts are increasingly being explored in meat products as potential natural antioxidants for improving oxidative stability due to the potent antioxidant potential of these compounds in the meat matrix. The antioxidant attributes of these plant extracts are due to flavonoids (flavanol, flavones, anthocyanins), phenolic acid (hydroxybenzoic, hydroxycinnamic acids), diterpenes tannins (hydrolyzable and condensed tannins), stilbenes, coumarins, lignans, quinones, curcuminoids, and others including phenolic alkaloids, phenolic glycosides, phenolic terpenoids, and essential oil) [4,5]. Tea phenolics, such as epigallocatechin gallate (EGCG), have antioxidant potential in terms of their having electron reduction (550 mV) potential comparable to alpha-tocopherols or vitamin E (480 mV) [6].

The antioxidant potential of these plant extracts can be measured by assessing their free radical-scavenging ability or by measuring the compounds associated with lipid and protein peroxidation. The free radical-scavenging ability of the plant extract is measured spectrophotometrically by using stable radicals, such as 1,1-diphenyl-2-picrylhydrazyl (DPPH) radical-scavenging activity or percent inhibition or 2-2-azinobis-3ethylbenthiazoline-6-sulphonic acid (ABTS) radical cation activity or percent inhibition, by measuring the ability of antioxidants to quench stable DPPH^+^ or ABTS^+^ radical cations, respectively, in ferric reducing antioxidant power (FRAP) by assessing the reduction of ferric-TPTZ (2,4,6-tri(2-pyridyl)-1,3,5-triazine (TPTZ) into ferrous from ferric thiocyanate assay (FTC), nitric oxide scavenging (NOS), aldehyde/carboxylic acid assay (ACA), superoxide scavenging, oxygen radical absorbance capacity (ORAC) and total antioxidant capacity (TAC). The antioxidant potential of the plant extract can also be measured by various compounds associated with lipid peroxidation (malonaldehyde formation, thiobarbituric acid reacting substances [TBARS], peroxide value [PV], free fatty acids [FFA]), protein oxidation (carbonyl formation, free thiols, conjugated diene) and beta carotene bleaching assay. Figure 1 depicts the antioxidant activity of the bioactive compounds extracted from plant biomass.

## 3. Extraction Protocols

The concentration of recovered bioactive molecules in extracts and their yield largely depend upon the various processing parameters. It is desirable to have a higher yield along with a significant concentration of active compounds. A proper extraction protocol is a primary step in ensuring the proper availability and utilization of these natural extracts, with higher preservative potential, and at an economical rate to industry [9].The overall composition of the extract obtained varies with the extraction protocols, plant species, part of the plant used (root, stem, leaves, root), geographical origin, harvesting time, type of storage, drying methods in case of use of powder and developmental stage of plant [10,11,12]. The use of extraction solvents, method of sample preparation, extraction duration, ratio of sample to extraction solvent, and temperature affect the composition and concentration of active compounds in an extract, thereby affecting its antioxidant efficacy [13,14,15,16]. Figure 2 depicts the various technical aspects of using plant extracts as a natural antioxidant in meat.

### Green Solvents for Extraction

Vegetable matter, such as specific parts of plants or as such a whole plant, when properly harvested for maximum yield, is cleaned and mostly dried at a low temperature to preserve the maximum activity of heat-labile active compounds. This dried material is ground into powder to increase its surface area for better segregation of bioactive compounds from the plant matrix. The finely ground powder is dissolved in either a suitable solvent or a combination of solvents (depending upon the suitability and efficiency of these solvents) and dissolves bioactive compounds from plant powders. The yield and concentration of active principles increased in the extracts upon increasing the total surface or contact area between the powders and solvents. In the plant kingdom, a huge variety of bioactive active ingredients showing antioxidant potential exist. These compounds have different chemical and physiological attributes that affect their solubility. Hence, these bioactive compounds are soluble in different solvents. It is practically not feasible to recommend a single universal extraction solvent to extract active ingredients from all plant biomass.

The commonly utilized solvents for extraction are ethanol, methanol, ethyl acetate, acetone, heptane, dimethyl sulfoxide (DMSO), and water. The food industry prefers to use solvents that are non-toxic and easy to handle. DMSO and methanol solvents for extraction have food safety issues due to their toxic nature; hence, mostly preferred as solvent of choice for various non-food-related uses. DMSO can be used for dissolving both polar and non-polar compounds, as well as its ability to mix in organic solutions, including water, completely. Methanol was also reported suitable for lower molecular weight polyphenols extraction whereas water has been recommended to extract polyphenols with larger flavanols [17]. Thus, a combination of these two has been recommended. Methanol retards polyphenol oxidase, and hence is recommended for phenolic-compounds extraction from the raw plant matrix [18]. Water is very efficient in extracting non-polar compounds, whereas the use of organic solvents proves efficient in extracting organic compounds from plant biomass. The hydrophilic compounds present in plants are the most active compounds that can be efficiently extracted by using water as a solvent. Water was also reported as a suitable extraction medium for tea catechins, as compared with combinations of methanol (80%) and ethanol (70%) [19]. Do et al. [20] reported significantly higher antioxidant activity in *Limnopholia* aromatic extracts obtained by absolute ethanol with maximum yield by the combination of 50% acetone and 50% water. They recommended using combinations of aqueous and organic solvents in suitable concentrations for the efficient extraction of active principles that are soluble in water and the organic solvent.

Organic solvents are commonly used to efficiently extract phenolic compounds, but their left-over residues may be potentially harmful to consumers’ health even in traces. Thus, while using organic solvents as an extraction medium, it is imperative to take utmost care to remove all extracting solvents from the filtrate. The application of aqueous media in extraction has several inherent advantages. Water is considered a cheap and the safest solvent but is not as efficient as organic solvents, especially in extracting active compounds possessing antioxidant potential [21]. Turkmen et al. [22] utilized water, acetone, N, N dimethylformamide, ethanol/methanol as extraction solvents for obtaining green tea and mate tea compounds and reported the use of 50% aqueous ethanol and 50% aqueous acetone to get the maximum yield and so antioxidant potential.

The food industry increasingly prefers green solvents for extraction due to their non-toxic nature, food safety, and recycling, such as water, ethanol, deep eutectic solvents, synthetic ionic liquids, and carbon dioxide, and water again, as a supercritical fluid in extraction [23,24,25]. The ionic liquids are salt mixtures in the liquid phase at room temperature, comprising ionic components that bind ionically [25]. These solvents have very low vapor pressure, high conductivity, are very stable at high temperature, and have a wide range of polarity, but their food-safety risk, high cost, and poor environmental degradability hampers their proper utilization in the extraction sector [26].

## 4. Extraction Methodology

Extraction conditions have a critical role in determining the overall efficacy of extracts. Generally, it is recommended to perform extraction at lower temperature and avoid prolonged exposure to high temperature, as the latter could lead to significant loss of active principle of the extract. Based on the type of plant biomass used, suitable solvent and extraction techniques must be employed to get the maximum concentration and yield of the active compounds in the extract. After extraction in solvents for a sufficient time, the solution is filtered. The resultant filtrate is dried at low temperature using advanced technologies such as vacuum (vacuum oven/rotator evaporator/rotavapor) or freeze-drying (for aqueous solvent mostly). Extracts should be stored under frozen conditions, in suitable, inert packaging material, such as glass bottles/trays, to prevent reaction between the active ingredients and packaging material. A solvent used in the extraction of bioactive compounds or essential oils should have low toxicity, a high mass transfer rate, a low boiling points, preserve the active ingredients, and should not interact with the organic compounds in the plant biomass or food [27].

### 4.1. Traditional Extraction Methods

Water or alcohol-based extraction methods, steam distillation, or Soxhlet extraction are widely used methods for extracting bioactive compounds from plant biomass. These methods are simple, low cost, and fast, but have some practical problems, such as difficult and complex operations, lower yield/inefficient methods, energy-intensiveness, and extract at high temperatures, leading to the degradation of heat-sensitive active compounds, problems in the purification of extracts, and removal of solvents—especially, poor recovery of solvents and the generation of organic wastes, a requirement of specific organic and inorganic solvents [28]. Under solvent extraction or portioning, active ingredients from plant biomass (solid or liquid) are separated by dissolving them into a suitable solvent. This is one of the most commonly used traditional methods by the food industry. The active ingredients (solutes/extracts) are extracted from one solid or liquid phase to another liquid i.e., solvent. This extraction process is generally completed at a higher temperature for long exposure, resulting in the possible degradation of the active principle in the extract and high energy consumption.

Under Soxhlet extraction, a small, dried sample is placed in an extractor and placed in direct contact with a suitable solvent (water, petroleum ether, and hexane) repeatedly until the extraction is complete. Its higher solvent consumption, energy requirement, and long extraction time are important issues with the Soxhlet extraction technology [29].

Maceration is the process of breaking/subdividing or softening matter into pieces of plant biomass in a suitable solvent. This method is recommended to preserve the typical essence of extracts of some valuable herbs that contain very delicate, heat sensitive, and volatile compounds. It is used for the preparation of traditional food products with specific flavors and organoleptic attributes.

Hydrodistillation is applied to extract volatile compounds from food/plants by using distilled water. This process, of extracting volatile organic compounds by azeotropic distillation and non-volatile organic compounds by boiling water, takes 6–8 h. It comprises three processes viz. penetration of water into solutes (hydrodiffusion), hydrolysis, and some level of degradation of heat-labile compounds due to high temperature [30]. Figure 3 provides a comparative description of conventional and green extraction technologies.

### 4.2. Greener/Advanced Extraction Methods

The high energy and solvent consumption and lower solvent recovery in traditional extraction methods has forced various researchers and biochemists to explore a more efficient method utilizing environmentally friendly and less harmful solvents. These technologies, such as supercritical fluid extraction (SFE), ultrasound-assisted extraction (UAE), subcritical water extraction (SWE), microwave-assisted extraction, instant controlled-pressure drop extraction (French-DIC *détente instantanée contrôlée*) are widely referred to as green extraction technologies [23,31]. These technologies require a lower amount of organic solvent or utilizeg less harmful or environmentally friendly solvents, show better reuse of solvents, use solvents from sustainable sources with reduced or no toxicity, consume less energy, have a lower environmental impact/footprint, better worker safety; that is, they are highly efficient extraction methods with higher yields, containing the maximum percentage of desirable bioactive ingredients in active form [23,32].

Various operational aspects of various green extraction technologies are described in Table 1.

## 5. Supercritical Fluid Extraction

A supercritical fluid is a dense gas or compressed liquid lacking any molecular interactions. These gases, in a supercritical fluid state, are compressed, exhibiting properties of both gas-like (mass transfer, diffusion, lower viscosity, higher penetration power) and liquid-like (high density, solvent power, reduced surface tension, and viscosity) attributes (mesophase) at temperatures and pressures above critical points, with even minor change/adjustment in these parameters resulting in a significant change in density (tunable). Active principle or bioactive compounds are separated from the plant matrix (solid or liquid) by utilizing supercritical fluids to extract solvents. Carbon dioxide and water are the two commonly used supercritical fluids in the industry on a large scale.

Recently, there is an increasing focus on extracting bioactive compounds from the plant by using various solvents such as carbon dioxide, propane, ethylene, and butane in the supercritical fluid stage (SCF), due to their use in efficient and environmentally friendly technology. This supercritical fluid technology has resulted in the development of several other subsidiary technologies, used in the pharmaceutical and food sectors, such as supercritical fluid extraction (SFE), gas antisolvent process (GAS), supercritical solutions (RESS), supercritical antisolvent process (SAS), and their suitable modifications, such as solvent-extraction systems (ASES), supercritical solvent impregnation (SSI), and supercritical assisted atomization (SAA) [34]. Supercritical fluid extraction (SCFE) is a fast, efficient, economical, selective, practically solvent-free method, having a simple procedure for sample preparations’ extraction of bioactive compounds from plant biomass [35].

### 5.1. Selection of SCF

#### 5.1.1. Carbon Dioxide as a Supercritical Fluid (SC CO_2_)

Carbon dioxide is the most commonly applied gas in SCFE. Supercritical fluid extraction with carbon dioxide is performed at 31 °C and 74-bar pressures. These processing conditions viz. temperature and pressure depend on the nature and type of the extracts to be obtained; for example, low pressure (100 bar) is used to extract volatile oils/essential oils, polyphenols, and unsaturated fatty acids [36]. SC CO_2_ is the most common (about more than 90% of total SCFs used) used supercritical fluid (SC CO_2_) used in the extraction of flavor and fragrance compounds. The SC CO_2_ has the following operational advantages:(a)Safe and user-friendly critical temperature (Tc-31.2 °C) and operates at low pressure (critical pressure Pc-7.3 Mpa) [34,37];(b)Low critical temperature, suitable for the extraction of heat-labile compounds;(c)© High density (467.6 kg/m^3^), at a critical point, leading to higher dissolving power;(d)Easily adjustable/tunable density, such as, at 42 °C, 766.5 kg/m^3^ density at 150 bar, 950 kg/m^3^ at 400 bar and 1075 kg/m^3^ (near to liquid CO_2_ i.e., 1256.7 kg/m^3^) at 750 bar. It allows the collecting of every compound present in the plant biomass by suitable processing conditions, such as [33]:(e)Readily available in the environment and economy;(f)Non-toxic, colorless, odorless, and non-inflammable gas;(g)Purity and recyclability;(h)Wide versatility during fractionalization and extraction.

Various parameters that make carbon dioxide as most common gas used for SCFE are presented in Figure 4.

The carbon dioxide gas in the supercritical fluid stage exhibits properties of a lipophilic solvent, with greater and easier manipulation of its solvent between its use as gas solvent versus a liquid one. SC CO_2_ has a low polarity index, and thus is useful in dissolving lipophilic compounds such as lipids and essential oils, but unsuitable for polar compounds. To overcome this challenge, SC CO_2_ is used with other co-solvents to increase its polarity index [38,39], or an anti-solvent with plant material is injected into SC CO_2_.

For the extraction of polar compounds, supercritical nitrous oxide (SC N_2_O) fluid (7.1 MPa Pc and 36.7 °C Tc) is utilized. Still, it has safety issues regarding its high concentration in organic compounds [40]. Water exhibits a supercritical fluid state at a high critical pressure (Pc 22.1 MPa) and critical temperature (374 °C Tc). Still, its corrosive nature has limited its use in the food and pharmaceutical industry [41]. The pre-treatment of plant biomass with water before extraction with SC CO_2_ or used as a co-solvent with carbon dioxide, in a subcritical or supercritical state, for the extraction of polar compounds from aromatic plants significantly improves the composition of its extracts [42,43]. Carbon dioxide can segregate flavor and aroma compounds from complex mixtures in various herbs and spices, such as ginger, garlic, and pepper.

Besides CO_2_, ethane (4.8 MPa Pc and 32.4 °C Tc), dimethyl ether, and propane (4.2 MPa Pc and 96.6 °C Tc) are also used in extracting bioactive compounds from plant biomass, due to their critical temperature and pressure being close to CO_2_ and better extraction efficiency in extracting polar compounds due to their higher polarizability as compared with CO_2_ [44]. There is a range of solvents that have the potential to be used in supercritical fluid extraction, such as n-butane, nitrous oxide, freon, propane, methoxymethane and water, but various food-safety matters and operational hazards/harsh handling should be given due consideration before their, application such as the very high critical temperature of water (400 °C), which poses serious handling challenges during its application in SFE.

#### 5.1.2. Propane as a Supercritical Fluid

Propane’s application in supercritical fluid extraction processes has certain inherent merits over carbon dioxide, such as higher yield, an improved solubility of non-polar compounds, and a fast and efficient process requiring less time with better specificity, and less solvent consumption [45]. Liquified petroleum gas (LPG) is a mixture of propane and butane used in domestic food preparation and heating. In extraction under high pressure (245 g/kg), it improves the operation speed. It improves the catalytic power of enzymes, such as increasing enzymes’ activity, leading to improved enzymatic hydrolysis of lignocellulosic material of sugarcane bagasse, for example [46]. The uses of propane and LPG as supercritical fluids in the extraction of bioactive compounds are presented in Table 2.

### 5.2. SCF Extraction Process

The SFE unit has the following essential components viz. a carbon dioxide compressor or pump, along with a modifier pump in a co-solvent, such as water or an organic solvent, an extraction reactor, and a collection or fractionation vessels. To improve the selectivity and sequential separation of active compounds from plant biomass in SFE operations, a series of reactors are applied, or a various combination of temperature, pressure, and solvent flow rate is applied [56]. Other processing variables that affect extraction in SFE are flow rate, extractor diameter, particle size (larger particles require more time), diameter-to-length ratio, the application of extractors, bed porosity, and stirring diameter [57]. Samples are pre-treated by various processes, such as microwaves, ultrasound, irradiation and enzymatic treatment, to increase the efficiency of the process and improve the recovery of bioactive compounds. The preparation of raw samples has an impact on the recovery of bioactive compounds, such as how air-drying may lead to a loss of heat-labile compounds, whereas freeze-drying helps in retaining the activity of these heat-labile compounds.

The optimization of processing conditions is critical for obtaining the desired outcome in supercritical fluid extraction. In general, an increase in pressure above certain points results in increasing fluid density and improving solute solubility, leading to higher yield. In some cases, pressure increase also proves disadvantageous, by decreasing the diffusivity of the supercritical solvents, thereby decreasing contact/interactions and dissolving solutes or even in some cases, cause the formation of void fractions and undesirable outcomes due to a compact solid matrix [58,59]. Temperature exhibits a crossover effect in SFE, where high-temperatures decreases yield by reducing density, increasing diffusivity, and imparting high energy to the system, whereas low temperature causes increasing extract yield by increasing density and reducing vapor-pressure solutes [33]. However, in essential oils, the diffusion has a greater impact than density, thus higher temperature result in higher yields [60].

Water is soluble in SC CO_2_ at 0.3% *v*/*v* [35], and this was observed to affect the extraction process, depending upon the nature of plant biomass. Paprika, for example, with 85% moisture recorded a marginal increase in yield [61], while no effect was observed on extract yield from *Nannochloropsis* sps containing 5–20% moisture [62], though extract yield increased (40%) and CO_2_ consumption was reduced by 25% by soaking the flowers of *Helichrysum italicum* prior to processing [63].

Overall, the SFE is completed in five basic steps:(a)Penetration of the matrix;(b)Supercritical solvent solubilizes the solutes/plant compounds inside the pores;(c)Internal diffusion of the solute until it has reached the external surface;(d)External diffusion of solutes from the solid–fluid interface of the supercritical fluid;(e)Precipitation of compounds/solutes by suitable pressure and temperature modifications [34,40].

The various components of an SFE unit are depicted in Figure 5.

### 5.3. Other Extraction Methods as SFE-Adjunct

The application of various other extraction methods in combination with SFE has been reported to improve the efficiency and recovery of active compounds in the SFE process. Porto et al. [64] observed an 11%-higher yield (up to 34%) than controls (23%) upon microwave pre-treatment followed by CO_2_ SFE at 40 °C and 300 bar with *Moringa oleifera* seeds at 100 W for 30 s.

#### 5.3.1. Enzyme-Assisted SFE

The enzymatic treatment of plant materials, such as cellulase, alpha-amylase, hemicellulose, and pectinase, also improves the extraction efficiency and yield by enzymatic degradation of the structural integrity of plant’s cell wall, thereby increasing the interactions between solvent and solute. Black pepper, treated with alpha-amylase obtained from *Bacillus licheniformis* and followed by CO_2_ supercritical fluid extraction (60 °C, 300 bar at 2 L/min flow) increased black pepper oleoresin yield by 53%, with a 46% more piperine-enriched extract and higher enzymatic activity, noticed in continuous flow (2.13%), than the batch process (1.5%) [65].

Pre-treating pomegranate peel with recombinant enzyme mixtures containing cellulase, pectinase, and protease (2:1:1) followed by SFE (using SC CO_2_ and ethanol as co-solvent) resulted in a two-fold increasing crude extract yield with higher levels of total phenolics as compared with extracts obtained by only the SFE process. Vanillic acid (108.36 μg/g), ferulic acid (75.19 μg/g), and syringic acid (88.24 μg/g) were reported as principal phenolic compounds in the pomegranate peel extract [66]. The authors recommended using enzyme-assisted supercritical fluid extraction (EASFE) as a state-of-the-art green technology for extracting bioactive compounds from plant biomass. The pre-treatment of black tea leftovers with 2.9% kemzyme (2.8% *w*/*w* at 45 °C, pH 5.4 for 98 min) followed by SFE using SC CO_2_, along with ethanol as a co-solvent, increased extract yield by five-fold, displaying a higher amount of caffeic and para-coumaric acid as compared with controls. The enzymatic treatment hydrolyzed the cellulose and hemicellulose of the black tea leftovers, thus releasing the non-extractable polyphenols [67]. The application of plant cell-wall glycosidase to freeze-dried tomato matrix, prior to SC CO_2_ (500 bar, 86 °C, 4 mL/min, SC CO_2_ flow) extraction, was observed to increase lycopene content three fold [68].

#### 5.3.2. Ultrasound-Assisted SFE

Ultrasound application at 10–20 kHz creates a mechanical breakdown of plant biomass and improves the extraction yield with less solvent and energy consumption. A longer ultrasound treatment, above permissible limits, causes undesirable outcomes due to non-enzymatic browning reactions and oxidative degradation, for example, a 200 W and 10 min ultrasonic treatment followed by SC CO_2_ extraction process increased oil yields by 3.3%, whereas further increasing the duration of ultrasonic treatment lead to an undesirable outcome [69]. Ultrasonic pre-treatment of ginger rhizome (40 kHz, 40 °C, 1 h), followed by SC CO_2_ extraction (250 bar, 40 °C and 15 g/min flow rate), resulted in a 110% improvement in extract yield as compared with controls [70]. The application of various extraction processes as SFE-adjuncts are presented in Table 3.

### 5.4. SFE of Bioactive Compounds

Various essential oils present in these aromatic herbs and spices are the principal compounds affecting the flavor and fragrance of food and pharmaceutical products, owing to the presence of oxygenated compounds such as phenols, ketones, aldehydes, acids, alcohols, ethers, acids, and esters, along with mono and sesquiterpenes hydrocarbons [34]. The flavor and aroma of food products has significant impact on its sensory attributes and acceptance by consumers. Various aromatic herbs or spices are added during their preparation to improve their flavor and aroma. These compounds also form a significant part of the global market of natural products. SFE is increasingly being applied in the food, pharmaceutical and cosmetic sectors as suitable alternatives to traditional solvent or steam distillation [35], the latter possessing lower selectivity, is energy-intensive/expansive, and exhibits a loss of volatile compounds during the extraction process [78]. The extraction of various bioactive compounds by using SFE technology are presented in Table 4.

## 6. Pressurized Liquid Extraction (PLE)

Under pressurized liquid extraction (PLE) or accelerated solvent extraction (ASE), solvents at high temperature (25–200 °C) and pressure (up to 200 bar or 20 MPa) are used for the extraction of bioactive compounds. The high temperature and pressure decrease the solvent’s surface tension, which facilitates penetration into the pores of the plant matrix, thereby improving the mass transfer of the active compounds to the solvent [127]. The solvent under pressure remains in a liquid state, even at its boiling point, and facilitates extraction at a higher temperature. Under these conditions, solvents that are not efficient in extracting analytes such as phenolic compounds or anthocyanins under normal conditions may be used for the same extraction. Pressurized solvents have improved, featuring desirable physicochemical properties, such as increased diffusivity, solubility, viscosity and dielectric constant, and these can be further modified by changing temperature and pressure. This is a rapid (completed within 30 min) and efficient process with reduced solvent consumption, but higher temperature-induced damage to heat-labile active compounds [128]. The requirement of large and sophisticated equipment and extraction at higher temperatures are drawbacks to this method.

The use of pressurized extraction technology for extracting bioactive compounds from plant biomass is presented Table 5.

## 7. Ultrasound-Assisted Extraction (UAE)

Plant biomass, exposed to high-intensity ultrasonic waves (20 kHz–100 MHz), results in tiny cavitations/bubbles around the cells. These bubbles suddenly collapse during the procedure, producing shockwaves that disintegrate the cell wall and release its intracellular contents, thereby improving the release of target compounds. This technology is simple to operate, has relatively inexpensive extraction systems, increases mass transfer, increases kinetics, yields bioactive compounds, and is suitable for a wide range of solvents for industrial applications [132]. There are two main physical phenomena in the UAE viz. diffusion of solvent through the cell wall and dissolving cell content after breaking the cell wall.

Ultrasound-treated plant biomass increases the extraction yield by fragmentation or reduction of particle size; erosion and improved accessibility to the solvent; sonocapillary effects, by increasing the velocity and depth of penetration of the solvent through pores and canals; sonoporation, by increasing cell membrane pore size; local shear stress by friction between the liquid molecules; detexturization; and combinations of these methods [133]. UAE efficiency is affected by the temperature of solvent, pressure, the frequency of ultrasound waves/energy, and the sonication time [134]. This method is relatively easy to use and can be suitable/applied for/to a range of plant matrices, features tunability, and requires low capital cost compared with other methods.

Ultrasound can be applied directly to the extraction medium by a probe, which amplifies the intensity of the ultrasound waves (up to 100 times), whereas indirect ultrasound treatment is given by using an ultrasound water bath, wherein the water acts as a transport medium for the ultrasound waves [135]. Ultrasound-assisted extractions of bioactive compounds from plant biomass are presented in Table 6.

## 8. Microwave-Assisted Extraction (MAE)

Microwaves (300 MHz–300 GHz frequency, 1 mm–1 m wavelength) penetrates food materials, agitating water molecules and charging ions, leading to non-contact heating. It enables faster heat transfer by converting electromagnetic energy to thermal energy through ionic conduction and dipole rotation [147]. This technique is commonly used to extract active ingredients, by combining microwave heating and traditional solvent-extraction techniques. This combination of microwave heating and conventional solvent extraction has several merits over conventional extraction methods, such as less use of solvents with better yields; it is afast process; it’s economics are better as it is a fast and efficient process with less energy consumption and carbon dioxide emissions [148]. Due to these procedural and technological advantages, this technology has become a popular method for extracting substances from plant materials.

At present, several advanced technological innovations have been incorporated in this method, such as pressurized microwave-assisted extraction (PMAE), and solvent-free microwave-assisted extraction (SFMAE). The microwave-assisted extraction resulted in a significant increase in apple-pomace-extract yield with active ingredients (55%), as compared with other extraction methods, such as ultrasound-assisted extraction (33%), pressurized liquid extraction (33%), and maceration (43%) [149]. Microwave-assisted extraction of bioactive compounds from plant biomass are presented in Table 7.

## 9. Pulsed Electric Field Assisted Extraction

Pulse electric field (PEF) is a novel, environment friendly, non-thermal and green technology. Its application increases the cell membrane permeability and mass-transfer rate. It has wide application in the meat industry, such as accelerated curing, drying and freezing, meat tenderization, novel product development, and restructured meat products. The application of PEF is performed by placing the food items between or passing it through two electrodes while applying a very high-voltage electric field for a very short duration [157]. To achieve a visible PEF effect on preservation and quality of food—the purpose here underconsideration—electric pulses of 20–1000 µs (several nanoseconds to several milliseconds) are required in an electric field several orders of magnitude strong (0.1–80 kV/cm) [158]. The overall efficiency of PEF depends upon the applied field strength, temperature, and energy-delivery efficiency. During the application of PEF, irreversible structural changes occur in cells’ membranes, leading to a marked increase in cell permeability and an enhanced mass-transfer rate across the membrane. These two factors ultimately result in the breakdown of cellular tissue [159].

This technology could be very useful in recovering bioactive compounds of more specificity from plant matrices more economically and with a low environmental impact, by softening and disrupting the cell membrane, resulting in the release of intracellular compounds [160]. Pulse electric field-assisted extraction can utilize renewable, alternative, efficient, and less toxic solvents obtained from plant resources, such as water and agri-solvents, which exhibit lower energy consumption, and produce higher quality and purity yields. Parniakov et al. [161] applied pulse field-assisted extraction of bioactive compounds from the *Agaricus bisporus* mushroom and obtained mushroom extracts with higher polysaccharide content and high purity with less energy. Boussetta et al. [162] also noted increased yield and improved extraction efficiency, even at low temperatures and with less solvent, in this method. It facilitates easier removal of bioactive compounds from the plant matrix without damaging it and facilitates segregation and purification later. Table 8 depicts the application of pulse electric field extraction of bioactive compounds from plant biomass.

Although PEF systems have low maintenance costs, high efficiency, and are fast, their exorbitant initial capital cost is still prohibitory. With the introduction of new technologies and the scaling-up of production, its cost is decreasing steadily and finding wider acceptance by the food industry.

## 10. Miscellaneous

The other extraction methods are high-voltage electric discharges and high hydrostatic pressure. The high-voltage electric current is applied through an aqueous solution through electrodes, inducing an avalanche of electrons, bubble cavitation, and pressure shock waves, causing cell damage and resulting in the release of bioactive compounds in a process similar to ultrasound [134,160]. The electric energy leads to chemical electrolysis and free radical formation, which may react with bioactive compounds, leading to reduced antioxidant activity of the extracts [161].

The high-hydrostatic-pressure method is seen as an alternative to thermal processes in improving the microbiological quality of food products. It is a green technology, involving thermal treatment and increased mass transfer rates and metabolites movement [168]. High-pressure treatment leads to protein denaturation by deprotonation and breakage of hydrophobic linkages and salt bonds, improving permeability and diffusion of the solvent, resulting in a higher yield of bioactive compounds [169].

Thus, the selectivity and yield of traditional and novel green extraction techniques depend on the proper selection of extraction protocols, such as critical input parameters, the nature and state of the plant biomass, the structure and heat sensitivity of its bioactive compounds, the yield of its extracts, and the equipment needed. However, these technologies should be commercialized and developed on a large scale, with suitable technological advancements and better, safer, and greener solvents to maximize benefits for the food, pharmaceutical, and nutraceutical sectors. Higher capital costs or larger initial investments are still prohibitive and must be reduced to facilitate their large-scale adoption.

## 11. Plant Extracts as Natural Antioxidants in Meat

The most common method of applying these extracts in meat processing is mixing them with water (mostly for water-soluble extracts; for organic solvent-assisted extracts, it is advised to mix in vegetable oil or fat) during preparation. It ensures the homogenous distribution of extracts in the product. The overall oxidative stability of products depends upon the concentration of the extracts; the higher the extract concentration, the greater the antioxidant effect. Phenolics, catechins, flavonoids, quercetin, anthocyanin, tocopherol, rutin, chlorogenic acid, lycopene, caffeic acid, ferulic acid, p-coumaric acid, vitamin C, protocatechuic acid, vitamin E, carotenoids, β-carotene, myricetin, kaempferol, chrysin, carnosine, zeaxanthin, sesamol, rosmarinic acid, chlorophyll, carnosic acid, carnosol, and gallic acid are compounds, present in plants, possessing antioxidant potential [170].

The important bioactive compounds present in various plant biomass and their meat application as natural antioxidants are described next.

*Moringa oleifera* a common vegetable in South Asian and African countries, is widely explored for its use as natural preservatives, owing to its various bioactive compounds viz. rhamnetin, isoquercitrin, kaempferol, kaempferitrin, saponins, triterpenoids, tannins, anthraquinones, alkaloids, and terpenoids [171], with concentration varying with the maturity of the plant and climatic and geographical conditions. *M. oleifera* is a rich source of protein, provitamins, vitamin C, A and E, zinc, calcium, iron, and potassium along with anti-cancerous agents such as glycerol-1-9-octadecanoate, glucosinolates, isothiocyanates, and glycoside compounds.

Rosemary contains a high amount of rosmarinic acid, carnosol, and carnosic acid in extract, and eucalyptol, α-pinene-bornyl acetate and camphor in rosemary essential oil [172,173]. Rosemary leaves were reported as a rich source of vitamin C, as much as as 18.51 g/100 g raw materials and extracts (0.26 mg/100 mL aqueous, 0.34 mg/100 mL alcoholic, and 0.36 mg/100 mL acetonic) have been reportedly obtained [10]. Monoterpenes hydrocarbons, esters, oxygenated sesquiterpenes, phenol, sesquiterpene hydrocarbons, oxygenated monoterpenes, ketones, and alcohol are the primary flavoring compounds of rosemary [173]. Generally, camphor, caryophyllene, borneol, bornyl acetate, and verbenone are chief compounds present in rosemary extracts. Peng et al. [174] advocated the application of supercritical fluids for the extraction of rosemary extracts, as it is associated with saving time, maximizing yield, and inhibiting the conversion of carnosic acid into carnosol. For rosemary stems, flowers, and leaves extracts, a suitable extraction methodology must be adopted, as lipophillic solvents and water (in the case of fresh samples were reported to result in a significant loss of phenols due to the action of phenoloxidase enzyme [175,176].

Arjuna or Arjun tree (*Terminalia arjuna*) bark, stem, leaves, roots, and fruits are a rich source of various beneficial bioactive compounds (polyphenols, flavonoids, triterpenoids, tannins, glycosides, sitosterol) and minerals. The Arjuna tree bark was reported to have the highest concentration of flavonoids such as arjunolone, quercetin, flavones, kaempferol, baicalein, and pelargonidin. Ethanolic and aqueous arjuna fruit extracts have been reported to contain a good amount of total phenolics (11.04–16.53 mg gallic acid equivalents/g), exerting significant scavenging activity (50.02–58.62%) [177].

Cinnamon bark is considered a promising antioxidant source, even exhibiting antioxidant potential comparable to synthetic antioxidants [178,179]. Most of the bark available in the market is Ceylon bark; dried bark (cork and cortex) of shoots of tree *Cinnamomum zeylanicum* F. *lauraceae*, containing approximately 0.5–1.0 % of volatile oil made up of 50.5% cinnamaldehyde, 8.7% cinnamic acid, methoxycinnamaldehyde and cinnamyl acetate, and 4.7% eugenol [178]. Chan et al. [180] reported ahigher antioxidant efficacy of deodorized cinnamon in meatballs, and noted significantly reduced lipid oxidation without causing any detrimental effects to its sensory attributes. Spices in the *Lamiaceae* family contain high rosmarinic acid, the major phenol (ranging from 1086–2563 mg/100 g of dry-weight) [181]. Rosmarinic acid possesses strong antioxidant activity due to two ortho-dihydroxy groups in its structure. This compound can act as an antioxidant and control the oxidation of low-density lipoproteins (LDLs). Rosmarinic acid also possesses anti-inflammatory activity by stimulating interleukin-10 (IL-10) secretion [182].

Several studies have confirmed the anti-inflammatory activity of oregano extracts, which are aqueous extracts observed to inhibit cyclooxygenase-2 (COX-2) secretion in epithelial carcinoma cells [183], and to exhibit anti-inflammatory properties by controlling stress-induced gastritis and hypersensitivity [184]. Oxidative stress leads to incorrect protein folding in the endoplasmic reticulum. Kaempferol, an aglycone flavonoid widely present in aloe vera, ivy gourd, saffron coccus and Peking spurge, is known to prevent hepatocellular carcinoma by controlling oxidative stress caused by reactive oxygen species [185]. Five bioflavonoids obtained from moss fern (*Selaginella doederleinii*) were observed to inhibit non-small cell lung cancer cells by suppressing XIAP and survivin expression, increasing the upregulation of caspase-3/cleaved-caspase-3, inducing cell apoptosis in A549 cells with low toxicity to non-cancer cells MRC-5 cells [186]. The *Kalmia angustifolia* extract exerted antioxidant, anti-inflammatory and anti-aging effects, at concentrations up to 200 μg/mL, by enhancing the expression of elastin and collagen-1 [187]. Flavonoids, such as apigenin, myricetin, and luteolin, were observed to exert anti-cancer effects against a range of human epithelial cancers by selectively reducing the viability of cancer cells, the alteration of ROS signaling, and the arrest of cell multiplication [188]. The extract of *Polyalthia* spp. is known to exert antioxidant, anti-ulcer, anti-plasmodial, anti-cancer, anti-microbial and anti-inflammatory effects due to the inhibition of COX-2 activity, inhibiting downstream prostaglandin E2 (PGE2) production, and inhibiting focal adhesion kinase, phosphoinositide 3-kinase, 3-hydroxy-3-methylglutaryl co-enzyme A reductase, and dipeptidyl peptidase 4 [189].

Clove exerts the most potent antioxidant activity among all spices and condiments commonly used in the food industry. Carnosic acid and carnosol, tocopherols, carotenoids, and sterols are important active ingredients present in herbs. Clove extracts mainly contain eugenol, caryophyllene, and eugenyl acetate. These two compounds have strong antioxidant potential and are considered the main bioactive compounds responsible for the increased antioxidant activity of clove, equivalent to vitamin E [190]. While processing, n-hexane solvent was recorded to produce a better yield, with high antioxidant activity (total flavonoid content- 15.54 mg GAE/g, total polyphenol content-54.05 mg GAE/g, FRAP = 0.69 mg/mL, DPPH = 0.29 mg/mL), and antimicrobial activity as compared with extracts obtained with other solvents (alcohol, water, and petroleum ether) [191]. Clove extract, at 0.25%, reported exerting antimicrobial activity with an 8.8 mm-to-9.27 mm minimum inhibitory concentration, as seen in *S. typhimurium* and *E. coli,* respectively [191].

Citrus byproducts/coproducts viz. peel and pulp are a rich source of various active ingredients, such as dietary fiber, minerals, vitamins, organic acids, flavonoids (ferulic, sinapic acids, and chlorogenic), phenolics (hesperetin, hesperidium, diosmin, and narirutin), carotenoids (carotene, zeaxanthin, lutein) [192,193,194,195]. These compounds are applied in the meat industry due to their associated health effects, such as antimicrobial, antioxidant, anticancer, anti-allergic, and antihypertensive effects [196]. The antioxidant activity of different citrus extracts varies with the methodology of their extraction, their fruit type, their environmental conditions, such as soil type and climate, their fruit-ripening stage, and their harvesting time [197,198]. Nayak et al. [199] reported the high antioxidant potential of *Citrus sinensis* peel extracts obtained by using microwave and ultrasound-assisted accelerated extraction technology at 337.16–433.09 mL/L DPPH (50% inhibition). Aqueous and methanolic extracts of lemon pomace had high antioxidant potential, wherein the methanolic extract exhibited higher antioxidant potential as compared with the aqueous extract, as measured in terms of DPPH assay (0.17, 0.13 mg Trolox equivalents/g, respectively) and ABTS (0.403, 0.458 mg Trolox equivalents/g, respectively) [200].

Catechin, present in green tea, is a group of flavonoids (flavan-3-ols), especially epicatechin, epicatechin-3-gallate, epigallocatechin (EGC), and epigallocatechin-3-gallate (EGCG), with EGC and EGCG regarded as the main active principle in green tea, and which is studied for its various antioxidant effects in food processing. The incorporation of tea catechins at 300 ppm in beef, duck, ostrich, pork, and chicken markedly reduced their TBARS values during refrigerated storage and was noted to exert 2–4 times higher antioxidant potential as compared with vitamin E [201]. Catechins reduced the production of putrescine and tyramine in dry, fermented pork sausage. Green tea catechins (GTC) and green coffee antioxidants (GCA) in linseed oil and fish oil, added in pork sausage at 200 mg/kg, decreased the lipid peroxidation of the sausages during seven days refridgerated storage. A significantly decreased lipid oxidation rate and higher organoleptic attributes were recorded in the fish- oil-substituted sausages [202].

Grape seed extract has a larege amount of polyphenols (such as epicatechin, gallic acid, resveratrol, and procyanidin dimers) and was reported to exert high antioxidant efficacy [203]. The antioxidant activity of grape seed is reported to be 20–50 times greater vitamins E and C [204], due to its possessing proanthocyanidins and oligomers of flavan-3-ol units. Grape seed extract (GSE) is the richest known source of polyphenolic compounds (catechins, flavanols, phenolic acids, anthocyanins, and proanthocyanidins) and has 20–50 times more antioxidant potential as compared with vitamins E and C, respectively [205,206]. Libera et al. [207] prepared grape seed extract by heating a grape seed slurry in water at 50–60 °C under high pressure (100–175 hPa or 10000–17500 bar). The authors evaluated the antioxidant potential of grape seed extract by incorporating it into pork neck containing *Lactobacillus rhamnosus* LOCK900 and reported that the lipid oxidation (PV-1% oleic acid and TBARS-0.46 mg MDA/kg) in the extract-incorporated samples was comparable to samples containing sodium ascorbate (PV-0.9% oleic acid and TBARS-0.53 mg MDA/kg).

The aqueous extract of *Cantharellus cibarius* hads high antioxidant potential due to its high concentrations of polyphenols, beta-carotene, and ascorbic acid, but has lower ABTS radical scavenging potency than vitamin C in cases where the extract is prepared by the water-decoction method i.e., first smashing and grinding, followed by boiling in water to obtain the extract [208,209]. The incorporation of *Cantharellus cibarius* water decoctions during the preparation of frankfurters, at 0.75- and 1.5% levels, resulted in significantly reducing the total plate counts with potent inhibitory action against *Candida albicans,* and improved the sensory attributes of the frankfurters during 60 days of storage under refrigerated conditions [209].

The ethanolic extract of mesquite leaf of *Proposis* was reported to exert high antioxidant potential due to its total phenolic content (278.5 mg GAE/g) and total flavonoid content (226.8 mg RE/g). The incorporation of the extract (0.05–0.10%) during the preparation of pork patties resulted in significantly improved oxidative stability of the treated patties as measured by theie marked reduction in TBARS value (90%) and conjugated dienes (40%) as compared with the positive control, i.e., patties with BHT [210]. The Carob (*Ceratonia siliqua* L.) tree is an underutilized tree of the Mediterranean region. Carob has high dietary fiber content and has 1.2–7.0% polyphenolic compounds, especially catechins, myricetin, rutin, and gallic acid [211].

The various plant extracts utilized in meat as natural antioxidants are presented in Table 9.

## 12. Current Scenario

The studied plant extracts showed strong antioxidant activity in meat products, owing to strong H°-donating activity, high radical-scavenging capacity or the ability to sequester metal catalysts. Various studies had been undertaken to standardize the extraction protocols and incorporation of extracts in meat products, and to document the preservative effect of plant extracts, as indicated, by reduced oxidation (in terms of lower TBARS value, PV, FFA value, carbonyl content, and free thiols) and microbial growth inhibition (in term of total plate count, coliform count, psychrophilic count, and food pathogens), inhibiting various carcinogens, reducing levels of heterocyclic aromatic amines, and better preservation/maintenance of sensory attributes (such as freshness, color, juiciness, flavor, and overall acceptability). At present, advanced technologies are being applied (such as supercritical water extraction, microwave-assisted extraction, ultrasonication, vacuum drying, and freeze-drying) to produce high-quality extracts with higher concentrations of active principle, on an industrial scale, for food application.

Various commercial preparations of some commonly used plant extracts are now readily available for application in food processing. The active ingredients or bioactive compounds from plant extracts are concentrated or purified using various ion-exchange adsorptions, followed by elution or chromatography technology. These are prepared using these improved and advanced technologies to obtain the maximum concentration of active ingredients in the extracts and protect any loss or degradation of active ingredients due to undesirable liquids, enzymes, and fermentation leading to alcohol production due to vinification. Angeletti and Sparapani [239] patented a process for the preparation of grape seed extract, containing a lower concentration of monomeric polyphenols and a higher concentration of procyanidin oligomers, by using a combination of acetone and methyl alcohol as a primary solvent in addition to ethyl acetate, methylene chloride, and dichloroethane.

The purified preparations of these plant extracts are now readily available in the market as premier products/dietary supplements/nutraceuticals. These are recommended for various health benefits, such as reducing oxidative stress, maintaining cardiovascular health, lowering LDLs and proper blood pressure, improving memory, controlling atherosclerosis, and anticancer effect. Natural Products Insiders Inc. California, USA had commercial preparation of grape seed extracts with the trade name ORAC-15 M (80% polyphenols and 15000 oxygen radical absorbing capacity) and recommended its use in alleviating the oxidative stress of the body. Other commercial preparations of Natural Products Insiders Inc. are MegaNatural, Enovita, and Leucoselect. Activin, Gravinol-S, Activin, Gravinol Super are some commercial preparations of grape seed extract produced and marketed by Inter Health Nutraceuticals Inc. (Broadfield Park, Crawley RH11 9RT, United Kingdom). Pycnogenol, an extract obtained from pine bark is commercialized by Natural health Science Inc. New York, USA and Horphag Research, Cointrin, Switzerland. It has a higher concentration of procyanidins, phenolic acids, and bioflavonoids, and is recommended for cardiovascular health, skin and eye care, proper brain functioning, and improving respiratory and women’s health. Applied Food Science Inc. has a wide range of commercial preparations of plant extracts, such as green coffee extract (GCE-50% chlorogenic acids from raw coffee beans), green coffee caffeine extract (Java G with 45% chlorogenic acid, 60% polyphenols, and 40% caffeine), organic ginger powder (PureGinger with 2% gingerol), cascara fruit extract (CoffeeNectar from coffee cherry), caffeine from green tea (Purtea), etc. Herbalox Seasoning HT-25 and Herbalox HT-25 are trade names of rosemary extracts manufactured by Kalsec Inc. Kalamazoo, Michigan, USA. Kemin Americas Inc., Iowa, USA, manufactures, who markets the rosemary extract as Fortium R 20.

The consumer preference towards healthier and greener alternatives to synthetic preservatives has been the main driving force of the rapid growth of plant extracts. The global market value of plant extracts was reported to be 43.32 billion USD in 2019 and is forecasted to grow with a compound annual growth rate (CAGR) of 6.0%. The market value of plant extract is expected to reach 66.81 billion USD by 2027 [240]. With the increasing preference of consumers towards green alternatives and minimally processed or natural foods, this sector is expected to maintain high growth in the near future.

## 13. Prospects and Challenges

As plant extracts are very potent antioxidants and rich in phenolic compounds, there is a high probability that, at higher concentrations or incorporations, these will lead to bitterness or aftertaste and darkened color of the developed product. It is always desirable to explore and use plant extracts, owing to their higher antioxidant ability with minimum effect on the sensory attributes of meat products. Alternatively, combinations of various plant extracts are also used to achieve desired effects without modifying/altering sensory attributes. As flavor and taste are crucial in determining the overall acceptability and acceptance of food products by consumers, these extracts should be added to meat products, if well within the recommended limit/optimum level. Another challenge before the meat industry is the differing outcomes of plant phenolics in in-vitro systems and in the food matrix. It has been noticed that these compounds exert a potent antioxidant effect, even at small concentrations, in in-vitro conditions, but need comparatively higher concentrations in meat model systems, which may result in bitterness and off-flavor. This could be due to interactions between these phenolic compounds with other meat components, such as muscle protein, lipid and minerals [241,242].

The application of plant extracts in the meat industry shows promising results in extending storage life and preserving product quality by significantly inhibiting the oxidation of meat lipids and protein and microbial growth. A proper extraction protocol is critical for obtaining the desired yield percentages and activities of plant extracts that determine their overall effectiveness and success. The high energy and solvent consumption, associated food-safety hazards due to organic solvent residue and waste, are major concerns among food technologists, to the extent that even brand extracts are being prepared from such green alternative methods [241]. The remnants of toxic solvent in plant extracts make them unsuitable for use in the food industry, so to for the loss of active ingredients due to improper high temperature and storage, change in color and flavor, bitterness, increasing cost of production are some important considerations and various food technologists, and researchers are studying these aspects. Further, the current focus of the food industry is on the proper segregation and identification of active principles and assessing their safety levels, toxicity, if any, and optimum levels of incorporation of these active principles in food.

Mbah et al. [243] and Porkorny [244] noted the economical cost, better safety, and easier application of synthetic antioxidants, compared with natural antioxidants, as major factors leading to the higher utilization of these synthetic antioxidants in the food industry. Consumer perceptions about stored products as stale, their preferences towards the use of meat products with shorter shelf lives, without preservatives and minimally processed are some factors limiting the use of these plant extracts in the meat industry. There is a need for exploring a better suitable, efficient, readily available, safer, easy-to-use application; proper labeling; and cost-effective materials for application in the meat industry. Proper and uniform legislation is required for extracts, as some extracts have been classified as safe for consumption since ancient times but nonetheless produce allergic reactions in some consumers upon consumption.

## 14. Conclusions

Natural extracts are extracted from various plant sources and are rich in several bioactive compounds possessing potent antioxidants. Some of these plant extracts exhibit comparable or even better antioxidant properties than commonly used synthetic antioxidants in the meat industry. Plant extracts should be utilized by following proper extraction protocols to obtain higher yields with higher concentrations of bioactive compounds exerting desired antioxidant effects. Green extraction technologies are proving to be a better and more efficient method for extracting bioactive compounds from plant biomass. However, the high initial cost and lack of scaling-up have proven to be a significant challenge for food technologists and biochemists. The application of combinations of extraction technologies is recommended, such as using conventional means with supercritical fluid extraction, and Soxhlet extraction paired with microwave-assisted extraction due to their synergistic effects. There is also a need for further identification and separation of the active ingredients in plant extracts and their potential toxicological effects and permissible limits in meat products.

## Authors Contribution

Conceptualization, A.Q.S., M.F.A.A., S.J., Writing original draft, A.M.A., P.K., Writing-review and editing, A.M.A., P.K., I.F.M.R., S.J., M.F.A.A., A.Q.S. All authors have read and agreed to the published version of the manuscript.

## Figures and Tables

**Figure 1 antioxidants-10-01465-f001:**
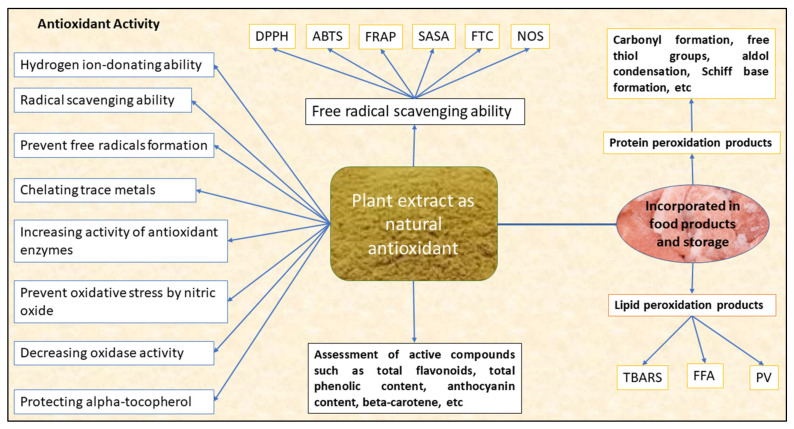
Overview of the antioxidant activity of the bioactive compounds extracted from a plant biomass source [7,8]. DPPH-1,1-diphenyl-2-picrylhydrazyl radical-scavenging activity, ABTS-2-2-azinobis-3ethylbenthiazoline-6-sulphonic acid radical cation activity, FRAP—ferric reducing antioxidant power, SASA-superoxide anion-scavenging ability, FTC—ferric thiocyanate assay, NOS—nitric oxide scavenging, TBARS-thiobarbituric acid-reacting substances, FFA—free fatty acids, PV—peroxide value.

**Figure 2 antioxidants-10-01465-f002:**
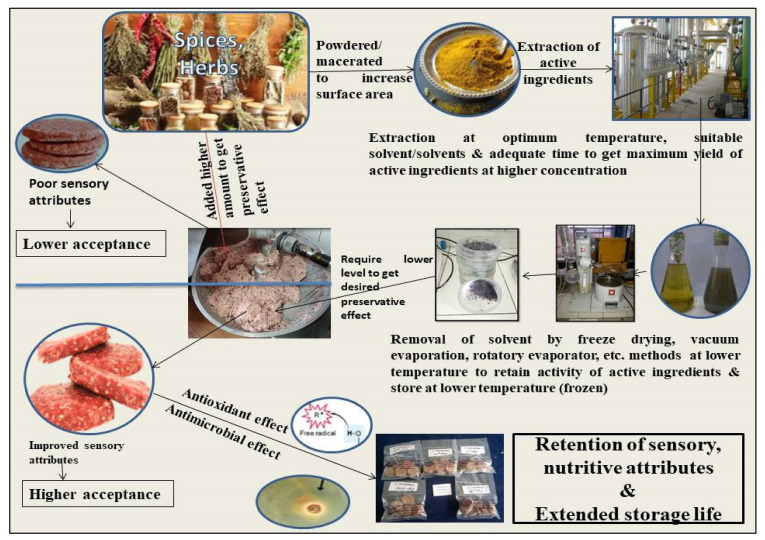
Plant extract as natural preservatives in meat.

**Figure 3 antioxidants-10-01465-f003:**
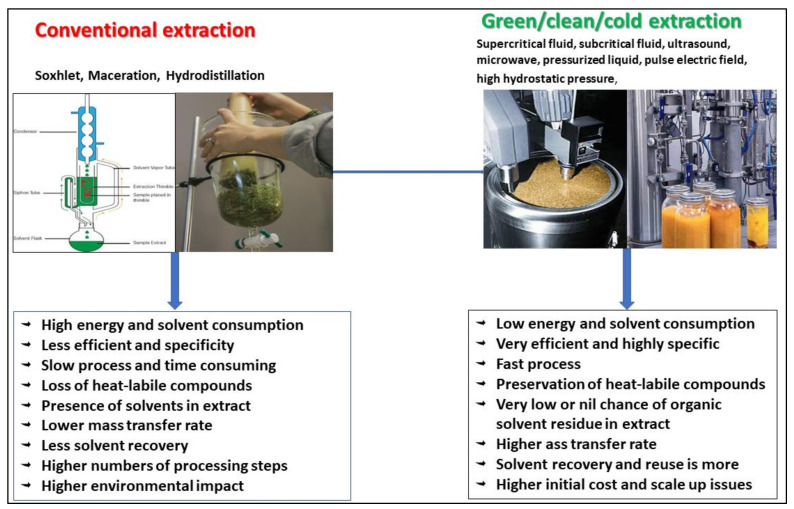
Comparative analysis of conventional and green extraction technologies.

**Figure 4 antioxidants-10-01465-f004:**
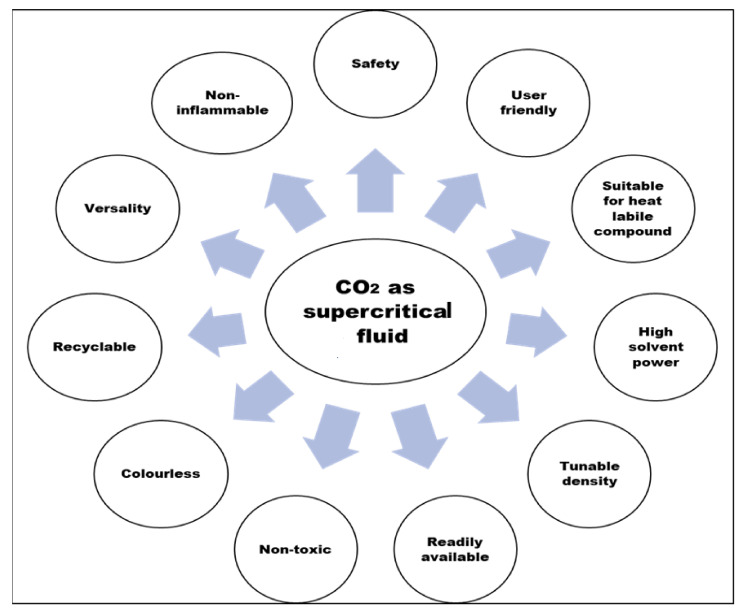
Suitability of CO_2_ in supercritical fluid extraction.

**Figure 5 antioxidants-10-01465-f005:**
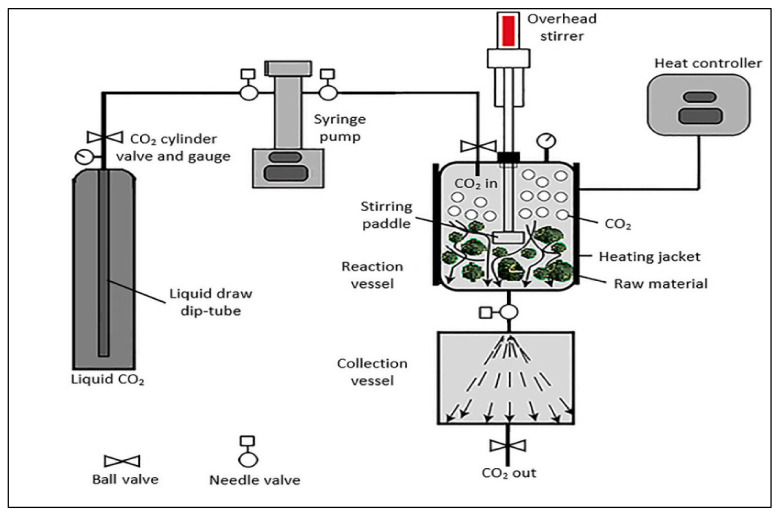
Supercritical fluid-extraction process. Adopted from [33].

**Table 1 antioxidants-10-01465-t001:** Technological aspects of various green extraction technologies.

Parameter	MAE	SFE	UAE	DIC
Process	Used along with traditional extraction methods to improve the extraction process	Very fast process	Used along with traditional extraction methods to improve extraction process	Rapid extraction
Solvent consumption	A small amount of solvent required	Very little amount of organic solvent or no solvent due to re-use	A small amount of solvent	Steam-driven progress with rapid depressurization
Solvent residue	Less solvent residue	No solvent residue due to phase separation on depressurization	Less solvent residue	Very low
Suitability/Applicability	Applicable for limited samples	Minimal application for a selected compound	High versatility and suitability	Used for sample pre-treatment process
Selectivity	Non-selectivity, extraction of a range of compounds	High selective for extraction of a small number of compounds	Non-selective extracts a range of compound	Non-selective extracts a range of compound
Processing conditions	High temperature and pressure	Not harsh conditions for SC CO_2_	Not harsh conditions	High temperature
Suitability for heat-labile compounds	Not suitable	Suitable to preserve the activity of heat-labile compounds	Suitable	Not suitable for heat-labile compounds
Energy consumption	High	Low due to re-use of solvent	Relatively low	High
Capital cost	Low initial capital cost	Very high	Lower capital cost	Very high
Technical workforce	Simple process	Needs very high technical workforce	Simple and easier operations	Needs high technical workforce

MAE—microwave-assisted extraction, SFE–supercritical fluid extraction, UAE–ultrasound assisted extraction, DIC—détente instantanée controlee, SC CO_2_—Carbon dioxide as supercritical fluid; source [33].

**Table 2 antioxidants-10-01465-t002:** Use of propane and LPG as supercritical fluid in the extraction of bioactive compounds.

Plant Material	Processing Protocol (Temp, Pressure, Flow Rate)	Remark	Reference
*Propane as supercritical fluid*			
Flaxseed	30–45 °C, 80–120 bar	28% higher yield of flaxseed oil with better composition, purity, and oxidative stability as compared with convention chloroform-methanol-water Soxhlet extraction	[47]
Perilla	40–80 °C, 80–160 bar, 1.0 cm^3^/min flow rate	Higher extraction yield of perilla oil with oxidative stability	[48]
Crambe seed	79.85 °C, 160 bar	The temperature has a vital role in affecting yield, less than 2% free fatty acids in the extract	[49]
Pequi pulp	30–40 °C, 50–150	43% higher yield of oil at 15 MPa	[50]
Canola seed	30–60 °C, 80–120 bar along with SC CO_2_ (40–60 °C and 200–250 bar)	Propane SFE faster than CO_2_ SFE	[51]
Sesame seed	30–40 °C, 20–120 bar along with SC CO_2_ (30–40 °C, 190–250 bar)	Extract quality same with both solvents, and temperature and pressure have an important role.	[52]
Sunflower seed	Propane and CO_2_	High concentration of tocopherol in the oil	[53]
*LPG as supercritical fluid*			
Rice bran	Compressed LPG and SC CO_2_	LPG decreased extraction time and save energy of re-compression	[54]
*Elaeis guineensis*	Compressed LPG	Advantages in terpene extraction with improve speed and reducing cost	[55]

SC CO_2_–supercritical carbon dioxide, LPG–liquified petroleum gas, SFE—supercritical fluid extraction.

**Table 3 antioxidants-10-01465-t003:** Application of various extraction processes as SFE-adjuncts.

Plant Source	Pre-Treatment	SC CO_2_ Protocols	Extract Yield	References
*Microwave-assisted SFE (MASFE)*				
*Moringa oleifera* seeds	100 W for 30 s	40 °C, 300 bar	11% higher yield	[64]
*Enzymatic-assisted SFE (EASFE)*				
Black pepper	amylase	60 °C, 300 bar, 2 L/min flow	53% increase yield with 46% higher piperine-enriched extract	[65]
Pomegranate peel	cellulase, pectinase and protease (2:1:1)	55 °C, 33 bar, 30–120 min, 2 g/min flow rate	vanillic acid (108.36 mu g/g), ferulic acid (75.19 mu g/g), and syringic acid (88.24 mu g/g) content in the extract	[66]
Black tea leftover	kemzyme (2.8% *w*/*w* at 45 °C, pH 5.4 for 98 min)	55 °C, 300 bar, 0.2–2 g/min flow rate, 30–120 min, ethanol as co-solvent	five-fold increase in extract yield	[67]
Tomato peel	glycosidase	500 bar, 86 °C, 4 mL/min	three-fold increase in lycopene yield	[68]
*Ultrasound-assisted SFE (UASFE)*				
Zinger	300 W, 20 kHz	40 °C, 160 bar, 4–8 mm particle size,	30% higher yield	[71]
Clove	185 W	32 °C, 95 bar, 0.233 *×* 10^−^^4^ flow rate, 115 min	11% higher clove oil with 1.2 times higher α-humelene	[72]
Korean perilla	750 W, 25 kHz for 125 s	25 °C, 100 bar, 1 h; ethanol as co-solvent	53% increase of luteolin and 144% increase of apigenin	[73]
*Capsicumbaccatum*	600 W	40 °C, 250 bar 1.7569 × 10^−4^ kg/s flow rate, 80 s	45% higher yield with 12% increase in capsaicinoid	[74]
*Capsicumfrutescens*	360 W	40 °C, 150 bar, 1.673 *×* 10^−4^ flow rate, 1 h	77% higher yield	[75]
*Hedyotis* *diffusa*	185 W	55 °C, 245 bar, 95 s	11–14% higher yield	[76]
Almond oil	20 kHz	55 °C, 280 bar, 55.6 *×* 10^−^^4^ flow rate, 510 s	24% higher yield	[77]

**Table 4 antioxidants-10-01465-t004:** Extraction of bioactive compounds from plant biomass by using supercritical fluids.

Plant Source	Extraction Medium	Extraction Protocols	Bioactive Compounds in the Extract	Remarks	Reference
Roasted peanuts	SC CO_2_	96 bar, 50 °C, fluid density-0.35 g/mL	74 flavor compounds (8–86 µg/kg) as hexanol, benzene acetaldehyde, methyl and ethyl pyrazines, methyl pyrrole, ethyl pyrazine, methyl pyrazines identified in the extract	Increasing roasting temperature and time significantly improved flavor compounds, with carboxylic acid becoming the most prominent	[38,39,79,80,81]
Coffee beans	SC CO_2_, 9.5% ethanol	200 bar, 100 °C	79% efficiency for acrylamide without affecting caffeine content in coffee	Temperature variation affected the extraction efficiency	[82]
Cumin	SC CO_2,_ toluene as static modifier	550 bar, 100 °C	Cumin essential oil concentration ranging from 1.74 to 3.51% (*v*/*w*)	Significantly decreased the extraction time from 8 h to 2 h	[83]
Turmeric	SC CO_2_, ethanol, isopropyl alcohol	In a fixed-bed extractor 300 bar, 30 °C	Increased curcuminoid content to 0.72% at 50% co-solvent without compromising extract yield percent (10%)	The best solvent mixture was 50% with 1.8 bed height/diameter ratio	[84]
Hyssop	Methanol (1.5% *v*/*v*)	101.3 bar, 55 °C, 30 min dynamic and 35 static time for sabinene	Sabinene (4.2–17.1%, *w*/*w*), iso-pinocamphone (0.9–16.5%), and pinocamphone (0.7–13.6%)	Composition of essential oil varies with extraction protocols	[85]
Black pepper	SC CO_2_	75–150 bar, 30–50 °C, particle size (0.5, 0.75 mm and whole berries)	Smaller particle size increase yield, Higher sesquiterpene concentration in SFE	Increase pressure and decrease temperature increase extract yield	[86]
Long pepper	SC CO_2_ with 10% ethanol, 10% methanol	400 bar, 40–70 °C	Piperovatine (0.93%) > palmitic acid > pentadecane > pipercallosidine	Drying leaves reduced amide concentration, the highest yield of piperovaltine by taking fresh leaves	[87]
Orange oil	SC CO_2_	131 bar, 35 °C, 2 kg/h flow rate	Increase concentration of oxygenated flavoring compounds (20 times more decanal)	Low temperature and flow rate improve fractionalization	[88]
Hop	CO_2_, ethanol, and water	111. 4 bar, 50 °C, 0.5 g/mL density	Highly concentrated oxygenated sesquiterpenoids	Reducing bitterness by decreasing lupulone and humulone	[89]
Eucalyptus globus	SC CO_2_, ethanol (0–0.5%)	200 bar, 40 °C	1.2% extraction yield, 50% concentration of triterpenic acid (5.1 g/kg of bark) with methyl 3-hydroxyolean-18-en-28-oate most abundant	About 80% more yield than conventional Soxhlet extraction method	[90]
*Polygala senega* and *Acorus tatarinowii*	SC CO_2_	450 bar, 35 °C, 2 h	24 compounds with 6 compounds (eugenol, beta-asarone, ethyl oleate, 1,2,3-trimethoxy-5(2-propenyl)-benzene, 6-octadecenoic acid, and 9–12-octadecadienoic acid) had more than 1.0%	Herb combinations increase the bioactive compounds with less compounds with one benzene ring compounds	[91,92]
Frankincense (*Boswellia carterii*)	SC CO_2_	200 bar, 55 °C, 94 min	The volatile oil contains 80% octyl acetate	SC CO_2_ extraction as the optimum extraction method	[93]
*Cannabis sativa* var. *indica*	Ultrasound extraction with cyclohexane and isopropanol solvent	100 bar, 35 °C, 1 mL/min	Isopropanol/cyclohexane 1:1 mixture, cycles 3 s, amplitude (80%) and sonication time (5 min) at 100 bar, 35 °C, 1 mL/min, no co-solvent for the terpenes and 20% of ethanol for the cannabinoids	Three monoterpenes and three cannabinoids were quantified in the ranges of 0.006–6.2 μg/g and 0.96–324 mg/g	[94]
*Croton zehntneri*	SFE CO_2_	66.7 bar, 15 °C	€-anethole α-muurolene, methyl chavicol or estragole, germacrene	Maximum solubility and yield at 20 °C	[95,96]
Bay laurel *Laurus nobilis berries*	SC CO_2_	90–250 bar 40 °C	€-β-ocimene (20.9%), α-pinene (4.2%),1,8-cineole (8.8%), β-longipinene (7.1%), α-bulnesene (3.5%)	15% yield, extraction at 250 bar produced an odorless liquid fraction with dominant triacylglycerols	[97]
Spearmint (*Mentha spicata*)	SC CO_2_, ethanol and ethyl acetate	50 °C and 300 bar	Carvone, 1,8-cineole, pulegone	Ethanol co-solvent has a maximum yield	[98]
	SC CO_2_	90 bar, 45 °C, 5 mL/s flow rate, 120 min dynamic time; 90 bar, 35 °C, 250 μm, 1 mL s^−1^, and 30 min	500 μm particle size has highest yield	2.03% extract yield and CO_2_ concentration 0.033 mg/mL	[99]
*Ocimum basilicum* (sweet basil)	Hydrodistillation and SC CO_2_	100–120 bar, 40–50 °C	Four times higher percentage of 1,8-cineole, 5–8 times, linalool, 1-2-fold eugenol, 28-fold germacrene	Higher t-cadinol and sesquiterpenes in essential oil	[100]
Clove basil (*O. gratissimum*	SC CO_2_	90–128 bar, 25–50 °C; 0.05–0.35 g/min flow rate	Eugenol (35–60%) and β-selinene (11.5–14.1%)	Solvent-to-feed ratio, 16:21; finely ground particle improves yield	[101]
Tomato skin and seed	SC CO_2_	300 bar, 60 °C, 0.16, 0.27, 3–8 h, 0.41 g/min flow rate	86% recovery of *E*-lycopen	Solvent to solid ratio 220 g CO_2_/g	[102]
Passion fruit bagasse	SC CO_2_	50–60 °C, 170–260 bar, 20.64 g/min flow rate	1.5- and 5.8-times higher tocopherols and carotenoids	SFE applied in the second stage improved the efficiency	[103]
Winter melon	SC CO_2_, ethanol	244 bar, 46 °C, 10 g/min flow rate, 97 min, 0.5 L extractor dimension	176 mg extract/g dried sample	Antioxidant activity of extract higher than obtained by ultrasound-assisted extraction or Soxhlet extraction	[104]
*Cannabis sativa*	SC CO_2_	300–400 bar, 40–60 °C, 1.94 kg/h flow rate	125.37 µg/g tocopherol in extract	2–3 times higher gamma-tocopherol content and higher alpha-tocopherol	[105]
*Carica papaya* fruit	SC CO_2_	200 bar, 80 °C, 16.45 mL/min flow rate, 3 h	Benzyl isothiocyanate (anthelmetic), carpaine and pseudocarpaine	Solvent/solid ratio 1180.4 g CO_2_/g	[106]
*Camelina sativa*	SC CO_2_	450 bar, 70 °C, 1 L/m flow rate, 510 min	Alpha-linoleic, oleic, eicosaenoic and erusic acids, higher phytosterol content	Solvent/Solid Ratio (gCO_2_/g)-16.14	[107]
Quinoa seed	SC CO_2_	185 bar, 130 °C, 0.175–0.45 g/min flow rate, 3 h, 1.2 mL extractor size	Four-fold increase in tocopherol content (336 mg/100 g oil) with SFE as compared with extraction with hexane	Solvent/Solid Ratio (gCO_2_/g)-8.02 to 67.5	[108]
*Crocus sativus*	SC CO_2_	349 bar, 44.9 °C, 10.1 L/h flow rate, 150.2 min	Extraction yield-10.94 g/kg with large amount of unsaturated fatty acid	Solvent/solid Ratio (gCO_2_/g) 1377.27	[109]
*Eugenia uniflora*	SC CO_2_ ethanol (polarity 5.2) and water (polarity-9.0)	400 bar, 60 °C, 2.4 g/min flow rate, 6 h	Trans-caryophyllene (14.18%), germacrenos bicyclogermacrene (40.75%), Selina epoxide (27.7%)	Solvent/solid Ratio (gCO_2_/g) 20.09 sequential extraction process most effective	[110]
*Moringa oleifera*	SC CO_2_	500 bar, 60 °C, 2 mL/min flow rate, 2 h	Selective extraction of 12 bioactive compounds in SFE	Solvent/Solid Ratio (gCO_2_/g) 37.85	[111]
		350 bar, 30 °C, 20 kg/h flow rate, 5 h, 2 L extractor diameter	Oleic acid (72.26–74.72%), sterol and tocopherol rich extract	Solvent/solid Ratio (gCO_2_/g) 1329.77	[112]
	Microwave pre-treatment (100 W, 30 s) followed by SC CO_2_	300 bar, 40 °C, 166.7 flow rate, 210 min, extractor dimensional 1 L	Microwave pre-treatment improves the extraction yield, polyunsaturated fatty acids, oil yield-35.28% *w*/*w*	Solvent/Solid Ratio (gCO_2_/g) 921.23–1000.2	[64]
*Pleurotus ostreatus*	SC CO_2_	210 bar, 48 °C, 333.33 g/min flow rate, 1.5 h, extractor dimension-100 mL	Phenol content: 5.48 mg GAE/g (dry weight) with 0.135 g dry weight content	Solvent/Solid Ratio (gCO_2_/g) 222,220	[113]
*Vine*/*Humulus lupulus*	SC CO_2_, ethanol, ethyl acetate and compressed propane	250 bar, 80 °C, compressed propane at 100 bar, 20 °C	Yield increases to 2.7% in compressed propane and 10.1% in SC CO_2_-ethyl acetate	Ethyl acetate as a co-solvent improve extraction yield and increases the concentration of bioactive compounds	[114]
*Catharanthus roseus*	SC CO_2_, ethanol	159 bar, Flow rate-0.3 mL/min, 8 min	Vincristine (size 5–200 nm) rich extract	Improve bioavailability	[115]
Cacao pod husk	SC CO_2_, ethanol (13.7%)	299 bar, 60 °C	0.52% extract yield having 12.97 mg GAE/g extract phenolic contents	Extract enriched in phenolic compounds, green technology	[116]
Yacon leaves	SC CO_2_, ethanol	250 bar, 70 °C, ethanol to solid ratio-3:1	High amount of total phenolic compounds and highest ω-6/ω-3 fatty acids ratios	Major unsaturated fatty acid in extract-gamma-linolenic acid, eicosapentaenoic acid and linoleic acid	[117]
Tilia flower	SC CO_2_, ethanol (5–10%)	220 bar, 65 °C, 15 min	Tiliroside as main flavonoids in the extract	Increase in temperature and pressure increase deficiency	[118]
*Odontonema strictum* leaves	SC CO_2_, ethanol	200 bar, 270 min	Three-fold increase in total flavonoid recovery containing 5 major flavonoids	The temperature does not affect extraction	[119]
*S* *age leaves*	SC CO_2_	150–200 bar, 25 °C, 90 min	High content of α-humulene, viridiflorol, and manool at low pressure (0.24–0.73%)	Pressure as the most critical parameter	[120]
*Piper leaves*	SC CO_2_, 5% methanol	220 bar, 80 °C	Germacrene D, pipercallosidine, 14-oxy-α-muuroleno, bicyclogermacrene and (*E*)-caryophyllene	40% more yield (1.36% to 2.18%) by using methanol co-solvent	[121]
*15 vegetable waste matrices*	SC CO_2_, 15.5% ethanol	350 bar, 59 °C, 15 g/min flow rate, 30 min	Total carotenoid recovery more than 90% with beta carotene dominant compound (88–100%)	SC CO_2_ valuable method for carotenoid extraction from vegetable waste	[122]
*Carrot peel*	SC CO_2_, 14.3% ethanol	58.5 °C, 306 bar, 30 min	5.31% yield having 96.2% higher carotenoid recovery	More manageable scale-up of the extraction process	[123]
*Rosemary*	SC CO_2_	3.4–172.4 bar, 40–50 °C, 600 μm particle size	Eucalyptol, camphor, and beta-caryophyllene as principal compounds	Essential oils yield—1.4–2.5 g/100g (*w*/*w*), with a higher yield than hydrodistillation	[124]
*Clove leaves*	SC CO_2_	220 bar, 40 °C	Eugenol (30%), chavicol (13%), n-pentacosane (12%), hexacosanal (11%), and vitamin E (9%)	High yield (1.8%) with eugenol as most prominent compound	[125]
*Radish leaves*	SC CO_2_, ethanol	400 bar, 35–40 °C	Total phenolic contents-1375–1455 mg GAE/100 g	Extracts exhibiting anti-inflammatory effects	[126]

SC CO_2_—supercritical carbon dioxide, GAE-gallic acid equivalent.

**Table 5 antioxidants-10-01465-t005:** Pressurized liquid extraction of bioactive compounds from plant biomass.

Plant Source	Solvent	Extraction Protocol	Extract Attributes	Reference
Jabuticaba skins	Ethanol	50 bar, 280 °C, 9 min	40-fold lower price, 2.15 times higher anthocyanin, and 1.66-fold higher phenolic content	[128]
Cranberry waste	Water, acidified water, ethanol, ethanol-water (50% *v*/*v*)	Ethanol and water at 100 °C	Total phenolic-7.36 mgGAE/g	[129]
Gooseberry	Water	16 bar, 52 °C, 51 min	11.68% yield of polysaccharides with high content of arabinose and glucose	[130]
Pepper	Water	200 bar, 120–240 °C, 10–20 min	113% higher extract yield as compared with conventional Soxhlet extraction	[131]

**Table 6 antioxidants-10-01465-t006:** Ultrasound-assisted extraction (UAE) of bioactive compounds from plant biomass.

Plant Source	Solvent	Extraction Protocol	Remark	Reference
*Eucommia oliver*	Hot water extraction followed by UAE	1:20 solid to liquid ratio, room temperature, 1 h	High yield with a higher concentration of natural antioxidants	[136]
Black chokeberry fruit	Ethanol (0–50%)-water	20–70 °C, 0–100 W, up to 4 h	High temperature and ethanol increased yield	[137]
Avaram shell	Distilled water with ultrasound probe	100 W with magnetic stirring (85 rpm), 5 h	1.6 times higher extraction of condensed tannin	[138]
Orange peel	4:1 ethanol-water	150 W, 40 °C	Increase yield of extract (11%) with higher polyphenols	[139]
*Moringa oleifera*	Ethanol: water (1:1)	40 °C, 15 min	Phenolic acids most prominent in extract	[140]
Artichoke residues	50% Ethanol	240 W, 10 min	95% higher retention of chlorogenic ac	[141]
Pine waste	Water	40 °C, 0.67 W/cm^2^ sonication intensity, 43 min	Pine sawdust as potential source of polyphenols (40% higher)	[142]
Pine seeds	Water with 0.2 N NaOH	25 °C, 30 KHz, 1 h	30% higher phenolic compounds	[143]
Flax seed	70–80% ethanol	30 °C, Material-solvent ratio of 0.55 g/mL	Higher content of azadirachtin	[144]
Piteguo fruit	Water	70 °C, 230 W, 13:1 mL/g solvent solute ratio	5.16% higher yield of extract	[145]
*Zizyphus* lotus fruit	50% ethanol	63 °C, 25 min, 67 mL/g solvent-solute ratio	High phenolic compounds (40.782 mg gallic acid equivalents/g dry matter) with higher antioxidant activity	[146]
Black mulberry fruit	Water	69 °C, 190 W, 40:25 solvent-solute ratio	Higher yield (3.13%)	[30]

**Table 7 antioxidants-10-01465-t007:** Microwave-assisted extraction of bioactive compounds from plant biomass.

Plant Source	MAE Protocols	Remark	Reference
*Terminalia bellerica*	100 °C, 40 mL/g, solvent-solid ratio in water	Maximum flavonoid yield (25.21 mg/g) with water with 82.74% recovery as compared with 63.75% in conventional methods	[150]
*Citrus unshiu* fruit peel	140 °C, 1 kW, 2.45 GHz, 8 min in 70% ethanol	47.7 mg/g hesperidin (86.8% higher yield)	[151]
Dragon fruit peel	100 W, 35 °C, 8 min	9 mg/L betalains (food additive, stable at broad pH and stable at low acidic food)	[152]
Chokeberries	300 W, 53.6% ethanol, 5 min	The highest yield of phenolic compounds (420.1 mg GAE/100 g)	[153]
Passion fruit skin peel	628 W, 9 min	Tartaric acid as best extracting agent for pectin, acetic acid, and nitric acid as agents for pectin extraction with better properties	[154]
*Citrullus lanatus* fruit rind	477 W, 128 s, solvent-solute ratio 1:20, 20.3 g/mL	Highest pectin yield (25.79%), hydrodiffusion microwave as the green and efficient extraction process	[155]
Boldo leaves	200 W, 56 min, 7.5% solid-solvent ratio	Efficient extraction of volatile and non-volatile organic compounds	[156]

**Table 8 antioxidants-10-01465-t008:** Pulse electric field (PEF)-assisted extraction of bioactive compounds from plant biomass.

Plant Source	PEF Extraction Protocol	Remark	Reference
Orange peel	60 μs (20 pulses of 3μs), 7 kV/cm	Improved naringin and hesperidin, total phenolic compounds increased up to 192%	[163]
Button mushroom	85 °C, 38.4 kV/cm	Increased yield of polysaccharides, phenolic compounds, and protein, a synergistic effect of temperature and electric pulses	[164]
Grape juices	1.5 kV/cm, electric conductivity-20 mS/cm, 50 Hz	Increasing anthocyanin, Vitamin C, and bioactive compounds having higher antioxidant potential	[165]
Borago leaves	300 Hz, 30 kV, 200 A current,	Polyphenol and antioxidant potential increased between 1.3 and 6.6-fold and from 2.0 to 13.7 fold, respectively as compared with conventional methods	[166]
Apple juice	3 μs, 3 kV/cm, electric conductivity-2.3 mS/cm	Higher polyphenols content and reduced processing time as compared with conventional methods	[167]

**Table 9 antioxidants-10-01465-t009:** Plant extracts as a natural antioxidant in meat.

Plant Source	Extraction Protocol	Experimental Design (Level, Meat Product, Storage Temp, Days)	Significant Outcome (Extract Quality and Its Antioxidant Effect on Incorporation in Meat)	Reference
Cinnamon barks	Ethanol (90%), 60 °C, 9 h	0.25%, chevon rolls, 4 ± 1 °C, 35 days	Overall acceptability of treated rolls was higher than control, significantly (*p* < 0.05) lower TBARS, FFA, PV, SPC, and psychrophilic count	[13,14,212]
Papaya leaves	Ethanol (60%), 65 °C, 15 min	0.5%, chevon emulsion, 4 ± 1 °C, 9 days	TBARS, FFA and PV (*p* < 0.05) higher in control than treatments	[213,214]
*Terminalia arjuna* bark	Ethanol (60%), 10 min at 75 °C	1.0%, pork emulsion, 4 ± 1 °C, 9 days	2.5-fold reduction in TBARS value than control (0.79 from 1.75 mg malonaldehyde/kg), better colour stability (*L *, a *, b ** values)	[215,216]
*Terminalia arjuna* fruit	Ethanol-water (60:40), 27 °C ± 1 °C, overnight, vortex shaking at 400 rpm for 8 h	1.0%, ground pork, 4 ± 1 °C, 9 days	Higher total phenolics (16.53 mg GAE/g), DPPH IC_50_—10.37 μg/mL, FRAP-1.33, Metmyoglobin content comparable to BHT added sample and significantly lower than control	[177]
*Oregano vulgare* leaves	Ethanol (60%), 80 °C, 10 min	1.0%, chevon emulsion, 4 ± 1 °C, 9 days	Total phenolic content-328.71 mg GAE/100 g, SASA-44.49%, DPPH activity-30.72%, improving oxidative and microbial quality of chevon meat	[213]
Clove buds	Ethanol (95%), 12 h at 100 rpm, residue again re-extracted	0.25%, 0.5%, 1.0%, 2.0%, Chinese-style sausage, 4 °C, 21 days	Concentration dependence effectiveness in controlling lipid and protein oxidation, better retention of textural and sensory attributes during storage	[217]
Watermelon rind	Ethanol (95%) 25 °C, 24 h at 200 rpm	0.10%, pork patties, 4 ± 1 °C, 28 days	DPPH (% inhibition)-77.46, ABTS (% inhibition)-75.57, FRAP (mM of Fe^++^ equivalent/mL)-77.5 and SASA (% inhibition)-47.5; zone of inhibition for *S. aureus*-5.68 mm	[218]
Sea buckthorn seeds	Methanol (60%), 55 °C, 20 min	0.30%, ground pork, 4 ± 1 °C, 9 days	TPC-128.23 mgGAE/g, DPPH-66.11% inhibition, ABTS-87.13% inhibition, significantly lower TBARS, FFA and PV in treated samples	[219]
*Moringa oleifera* leaves	Water for 18–20 h at 40–50 °C	0.10%, goat meat patties, 4 ± 1 °C, 15 days	TPC-48.36 mgGAE/g, TFC-31.42 mg/g, Lower TBARS value on 15 th day of storage in treated sample-0.53 mg malonaldehyde/kg	[220]
	Boiled distilled water, 5 min	450–600 ppm, raw and cooked patties	TPC-60.78–70.27 mg/g, non-significant reduction in metmyoglobin formation in control and treated samples during storage	[220,221]
Ginger rhizomes, potato peel, seeds of fenugreek	Ethanol (90%), room temperature, 1 h at freeze dried −60 °C	500–1000 ppm, ground beef patties, 5, 25 & 37 °C, 12 days	Ginger rhizome extract has the highest antioxidant (% inhibition)-(77.4) followed by fenugreek seeds (71.4) and potato peel (59.5)	[222]
Garlic ginger and onion	Water, 40 °C,30 min, Ultrasonic extractor (200 W, 40 kHz)	5–10% ginger-garlic-onion, stewed pork, 4 °C, 12 days	Synergistic effect of combinations of extracts, storage life extended to 5–6 days	[223]
Leaves of hyssop and rosemary	Dimethyl sulfoxide for 5 h at ambient temperature	Solution with 5.8 pH, cooked pork meat, 4 °C, 8 days	Hyssop and rosemary extract inhibit lipid oxidation and metmyoglobin formation	[224]
Leaves of myrtle, lemon balm, rosemary and nettle	De-ionized water ambient temperature, 15 min	10% each extract, ground beef, 20 ± 2 °C, 120 days	Inhibited lipid oxidation (lemon and nettle-23–24% lower peroxide value; myrtle and rosemary-33–41%) and protected colour	[225]
Green tea and grape seed	Boiling water, 10 min	500, 3000, 6000 ppm, Baladi goat meat, 5 °C, 9 days	Lower antioxidant capacity of green tea extract (7.5 h) than grape seed extract (9.4 h), plant extract increased the induction time	[226]
Red grape pomace	Methanol ambient temp, 10 min, sudden pressure changes to 5 × 10^3^ Pa (N/m^2^), rotatory evaporator at 200 rpm at 50 °C	0.06 g/100 g, pork burger, 4 °C, 6 days	TPC-546.0, total anthocyanins-1783.5 mg/L, antioxidant capacity-141.8 mmol/L Trolox, the application of instantaneous high-low pressure increased the extract yield	[227]
Wine residues	Aqueous acetone (50%), ambient temperature	7–15 g/100 g, dried minced pork slice, room temperature, 21 days	Decreased hexanal, TBARS (up to 108%), carbonyls, sulfhydryl loss	[179]
Mustard leave *kimchi*	Ethanol (70%), room temperature, overnight	0.05%, 0.1% & 0.2%, ground pork, 4 °C, 14 days	Extract at 0.1% and 0.2% having antioxidant effect equal to 0.02% ascorbic acid. MDA concentration below 0.5 mg/kg at the end of storage	[228]
Lotus rhizome knot (LRK) and leaf (LL)	Aqueous, room temperature, overnight	3%, bovine and porcine meat, 4 °C, 10 days	TPC-(LRK-17.0 gGAE/100 g, LL-34.9 g GAE/100 g), TTC-(LRK-13.02 gGAE/100 g, LL-6.02 gGAE/100 g), TFC-(LRK-7.96 g rutin euivalent/100 g, LL-33.0 g rutin equivalent/100 g)	[229]
Curry berry	Boiled water for 2 h followed by centrifuge at 5000 rpm for 10 min	2.5–5.0%, raw chicken meat homogenate, 4 ± 1 °C, 48 days	TPC-9.5 mg TAE/gdw, TFC-11.9 mgCE/gdw; the extract incorporation inhibited oxidative changes in meat	[230]
Lychee fruit pericarp	Boiled distilled water, 1 h	0.50, 1.0 and 1.5%, sheep meat nuggets, 4 ± 1 °C, 12 days	TPC-18.36 mgGAE/g, high anthocyanins content, the extract has good antioxidant potential.	[231]
Byproducts of olive, pomegranate, tomato and grape	Water, 60 °C, 2.5 h	0.1%, lamb patties, 4 ± 1 °C, 7 days	Water extracts exhibited antimicrobial and antioxidant potential, red grape and olive extract (1000 mg/kg) in patties reduced microbial counts	[232]
Bamboo shoot	Boiled water with 1% NaCl, 10 min	6% kordoi juice and 4% aqueous extract, pork nuggets, 4 ± 1 °C, 35 days	TPC-246 mg GAC/100 g, Ascorbic acid-4.1 mg AAE/100 g, The incorporation of extract and kordoi juice extended storage life from 21 days to 35 days	[233]
Colombian berry	Ethanol-water (50:50 *v*/*v*), solvent-solute ratio (5:1), 4 °C, lyophilized (0.18 bar, −50 °C)	250, 500 and 750 ppm, pork patties, 2 ± 1 °C, 9 days, 15–20 lux value	TPC-83976 mg/kg, total anthocyanin content-29077.5 mg/kg, making upto 35%. Extract improved colour stability and oxidative stability in dose dependent manner.	[234]
Petals blue pea flower	Spray-dried, vacuum packaged	0.02–0.16% *w*/*w*, pork patties, 4 ± 1 °C, 12 days	TPC-28.8 mgGAE/g, TEAC value of cooked patties-0.10–0.167 mg TE/g; Addition of 0.16% extract protect lipid and protein oxidation during storage	[235]
Bee pollen	Ethanol, 40 °C, 1 h, 150 rpm, lyophilized	0.02%, pork sausage, 4 ± 1 °C, 30 days	TPC-19.69 mgGAE/g, 10 mg/mL can neutralise 91.93% of beta carotene.	[236]
Monkfruit	Water, 200 W ultrasound power, 80 °C, 2 h	7–15 g/100 g, dried minced pork	98.51% DPPH inhibition at 200 g/L, 34.93% mongroside in extract. Extract delayed hexanal formation, TBARS, carbonyls and sulphydryl loss	[179]
Jabuticaba	Water, 60 °C, 6 h, microencapsulated	2–4%, fresh pork sausage	TPC-15.63 mg GAE/mg, FRAP-20.51μmol equivalent Trolox/g, Extract added fresh sausage as natural colorant had an antimicrobial and antioxidant effect	[237]
Peanut skin	Ethanol (80%), 60 °C, 50 min; followed by 15 min sonication at ambient temperature	3.0%, chicken patties 1 ± 1 °C, 15 days	TPC-32.6 mg GAE/g, FRAP-of 26.5 μmol Trolox equivalent/g. Decreased a * values (*p* < 0.05) and reduced lipid oxidation, with 0.97 malondialdehyde (MDA)/kg as compared with 19 mg MDA/kg	[238]

TBARS-thiobarbituric acid reactive substances, FFA-free fatty acid, PV-peroxide vale, DPPH—1,1-diphenyl-2-picrylhydrazyl, ABTS-2-2-azinobis-3ethylbenthiazoline-6-sulphonic acid, GAE–gallic acid equivalent, SPC–standard plate count, MDA–malondialdehyde, FRAP—ferric reducing antioxidant power, SASA–superoxide anion radical scavenging assay, TPC–total flavonoid content, TFC–total flavonoid content, TTC–total tannin content, TAE—tannin acid equivalent, AAE—ascorbic acid equivalent, TEAC value–Trolox equivalence antioxidant capacity, * *p* < 0.05.

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
