# Peer review of "Green Extraction of Bioactive Compounds from Plant Biomass and Their Application in Meat as Natural Antioxidant"

_antioxidants, 2021, doi:10.3390/antiox10091465_

Round 1
Reviewer 1 Report
Dear Authors, The review "Green Extraction of Bioactive Compounds from Plant Biomass and Their Application in Meat as Natural Antioxidant" is very interesting and innovative, but it lacks the part concerning the biological properties of the extracts obtained by various methods. Few examples have been reported in this review (for example lane 650-653 in miscellaneous) and in my opinion these studies have not been sufficiently exhaustive. Therefore, I recommend the authors to make a paragraph dedicated to recent papers, which describe in vivo biological properties of extracts obtained with the main techniques described in this review.
Author Response
Reviewer Comment-
Dear Authors, the review "Green Extraction of Bioactive Compounds from Plant Biomass and Their Application in Meat
as Natural Antioxidant" is very interesting and innovative, but it lacks the part concerning the biological properties of the extracts obtained by various methods. Few examples have been reported in this review (for example lane 650-653 in miscellaneous) and in my opinion these studies have not been sufficiently exhaustive.
Therefore, I recommend the authors to make a paragraph dedicated to recent papers, which describe in vivo biological properties of extracts obtained with the main techniques described in this review.
Authors Response:
We appreciate the encouraging words by the honorable reviewer. As per critical observations made by the Reviewer, we authors have added a paragraph comprises recent literatures on the biological activity of plant extracts i.e., line 590-610 as-
Several studies confirmed the anti-inflammatory activity of oregano extracts, aqueous extract observed to inhibit cyclooxygenase-2 (COX-2) secretion in epithelial carcinoma cells [184], and anti-inflammatory properties by controlling stress-induced gastritis and hypersensitivity [185]. The oxidative stress leads to incorrect protein folding in endoplasmic reticulum. Kaempferol, aglycone flavonoid widely present in aloe vera, Ivy gourd, saffron coccus and Peking spurge, is known to prevent hepatocellular carcinoma by controlling oxidative stress caused by reactive oxygen species [186]. Five bioflavonoids obtained from moss fern (Selaginella doederleinii) was observed to inhibit the non-small cell lung cancer cells by suppressing XIAP and surviving expression, increasing upregulation of caspase-3/ cleaved-caspase-3, inducing cell apoptosis in A549 cells with low toxicity to non-cancer cells MRC-5 cells [187]. The Kalmia angustifolia extract exerted antioxidant, anti-inflammatory, and anti-aging effects at a concentration up to 200 μg/mL by enhancing the expression of elastin and collagen -1[188]. Flavonoids such as apigenin, myricetin, and luteolin was observed to exert anti-cancer effect against a range of human epithelial cancers by selectively reducing the viability of cancer cells, alteration of ROS signaling and arrest of cell multiplication [189]. The extract of Polyalthia spp was known to exert antioxidant, anti-ulcer, anti-plasmodial, anti-cancer, anti-microbial, and anti-inflammatory effects due to inhibition of COX-2 activity, inhibiting downstream prostaglandin E2 (PGE2) production, inhibiting focal adhesion kinase, phosphoinositide 3-kinase, 3-hydroxy-3-methylglutaryl co-enzyme A reductase and dipeptidyl peptidase 4 [190].

Reviewer 2 Report
The current review on green extraction of natural antioxidants from plant biomass for application in the meat industry is very interesting and aligns with the goal, scope, and audience of Antioxidants.
The paper is well-written and provides a comprehensive overview of the subject. However, I found it to be too long and, at times, repetitive. As a result, it is difficult to keep the reader's attention on the purpose of this review: the application of green extraction in meat industry.
In particular, although interesting, the section on plant extraction (paragraphs 3-10) is in my opinion too long, detailed, and not always necessary in relation to the rest of this review, which deals with their use as natural antioxidants in meat. It would be preferable to condense it and include only relevant references to the targeted application.
As written, it appears that there are at least two different reviews with which it is sometimes difficult to make the connection. When we get to the meat application, we sometimes lose compounds whose extraction looked interesting in the "green extraction" section, because this application was clearly not targeted by the cited authors.
There are also some typos that will be simple to fix, especially during the final proofreading with MDPI's excellent proofreading service. But worryingly, there are occasionally mismatches between the references and their numbers in the text (as for example in Table 6 where there is a shift of a citation from 156).
I recommend the Authors revise their work accordingly before publication in Antioxidants.
Author Response
Reviewer's comment: The current review on green extraction of natural antioxidants from plant biomass for application in the meat industry is very interesting and aligns with the goal, scope, and audience of Antioxidants. The paper is well-written and provides a comprehensive overview of the subject.
Authors' response: We express our sincere gratitude for the inspiring and motivational words.
Reviewer's comment: However, I found it to be too long and, at times, repetitive. As a result, it is difficult to keep the reader's attention on the purpose of this review: the application of green extraction in meat industry. In particular, although interesting, the section on plant extraction (paragraphs 3-10) is in my opinion too long, detailed, and not always necessary in relation to the rest of this review, which deals with their use as natural antioxidants in meat. It would be preferable to condense it and include only relevant references to the targeted application.
Authors' response: We have shortened the section on plant extraction (paragraphs 3-10) into three paragraphs as per suggestion and valuable observation made by the Reviewer by focusing only on the natural antioxidants
Reviewer's Comment: As written, it appears that there are at least two different reviews with which it is sometimes difficult to make the connection. When we get to the meat application, we sometimes lose compounds whose extraction looked interesting in the "green extraction" section, because this application was clearly not targeted by the cited authors.
Authors' response: We appreciate the observations of the Reviewer, and thus, have connected the extracts obtained by green extraction technology and their application in the meat sector. With the popularization and scale-up of these technologies, the extracts prepared from these advanced methodologies will be used in the meat sector on a large scale in the near future.
Reviewer's comment: There are also some typos that will be simple to fix, especially during the final proofreading with MDPI's excellent proofreading service. But worryingly, there are occasionally mismatches between the references and their numbers in the text (as for example in Table 6 where there is a shift of a citation from 156).
Authors' response: We have corrected the references and confirmed their proper citations in the text and bibliography section.

Round 2
Reviewer 1 Report
I thank the authors for answering my questions and including the bibliography